# Shape-transitions of a morphing illusory contour can be decoded during multiple-object tracking from the ongoing EEG

Christian Merkel [1,2] ✉, Matthias Merkel [3], Jens-Max Hopf [1,2] & Mircea Ariel Schoenfeld[1,2,4]

The human visual system continuously extracts a wealth of dynamic information from the incoming retinal signal. One important task is the simultaneous tracking of multiple items that are moving within the visual environment. Past work has proposed that such multiple-object tracking relies on attentional resources that are location-based, i.e. resources are respectively associated with the individual items spatial positions. However, another possibility is that attentional resources are object-based, i.e. allocated to the combination of all items as an abstract shape configuration. Indeed, recent data suggests that during multiple-object tracking, the visual system continuously maintains a configuration represented by an illusory polygon formed by the shortest closed path connecting all tracked items. Here, we test this hypothesis by comparing signatures in the electroencephalographic (EEG) signal of 38 subjects with topological transitions of the polygon's shape. The topological transitions are qualitative changes in the polygon shape that go beyond mere quantitative changes in item positions (i.e., polygon corners). During object motion, the shapeshifting polygon can display switches between a convex and a concave shape. Concave shortest-path polygons can exhibit changes in the order in which the tracked objects are connected ('flips'). We demonstrate that topological transition events can be decoded from the ongoing EEG signal, revealing how the abstract configuration and its qualitative changes are represented throughout the tracking. Taken together, our work demonstrates that an object-based attentional mechanism is crucial during multiple-object tracking.

Active tracking of several visual items within a dynamic scene is an indispensable feature of the visual system. Yet, it is still subject to ongoing debate how multiple items can be attended to simultaneously, despite limited cognitive resources.

The individual capacity to attend multiple moving items has been measured using the multiple-object-tracking (MOT) task[1], which requires subjects to track a number of relevant target items among a set of irrelevant distractor items over a period of time. The subject is then presented with a probe set of items, and has to report whether this set exclusively consists of target items (Fig. 1a). Many studies confirmed that task difficulty is modulated as a function of item speeds, inter-item distances, or number of tracked items[2–6]. These results suggest that tracking is limited by finite cognitive resources. This is commonly interpreted in terms of a location-based

model of resource allocation, where items are tracked by maintaining their spatial positions individually[1,7].

Over 30 years ago, Yantis[8] suggested that subjects' performances during MOT tasks may be additionally facilitated by an object-based attentional process, which operates on the configuration defined by the whole set of target items. Yet, behavioral, electrophysiological, and functional brain imaging evidence for such an object-based mechanism has been put forth only recently[9–11]. In contrast to previous experiments, this work probed sets of items which were composed of targets and distractors. For one, the behavioral performance showed a parametric decline with increasing number of probed targets, confirming location-based accounts of tracking. Strikingly, however, when all target items were included in the probe set, a distinct behavioral advantage over the other conditions was observed. Furthermore, a corresponding electrophysiological signature was

[1]Department of Neurology, Otto-von-Guericke University, Magdeburg, Germany. [2]Leibniz-Institute for Neurobiology, Magdeburg, Germany. [3]Turing Center for Living Systems, CNRS, CPT, Aix Marseille Univ, Université de Toulon, Marseille, France. [4]Kliniken Schmieder, Heidelberg, Germany. ✉e-mail: christian.merkel@med.ovgu.de

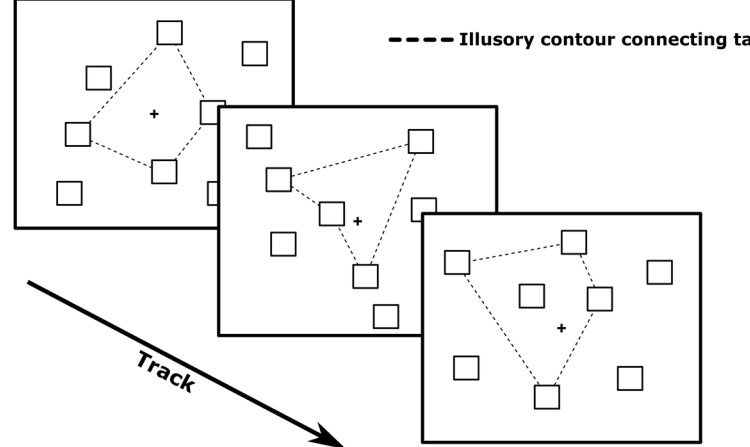

### a) Object-Tracking task

### b) Continuous morph of illusory contour

### c) Topological transitions

- ——— flip event
- ——— concave event
- ——— convex event

**Fig. 1 | Object-tracking paradigm. a** Subjects were required to attend four out of eight items. Subsequently, all items moved pseudorandomly across the screen for 4 s while subjects had to keep track of the four target items. At the end of each trial, subjects had to indicate whether four highlighted (probed) items were identical to the four target items or not by button press. **b** The positions of the individual items keep moving throughout the tracking phase. The dashed line depicts the shortest path connecting the four target objects. This path is unambiguously defined for each frame but was never visually present on the screen. **c** The continuous morph of this illusory contour occasionally underwent three topological transition: (blue) A "flip" was defined by a discontinuous morph of the illusory contour caused by the sudden change of the polygon sequence defining the shortest non-intersecting path; (red) The transition from a convex to a concave shape; (green) The transition from a concave to a convex shape.

observed, exhibiting an exclusive negative enhancement around 170 ms in response to probes consisting of the entire target set. These distinct response patterns suggest that during MOT, object-based mechanisms operate in addition to location-based ones.

Additional evidence for an object-based mechanism comes from a study where we probed the spatial distribution of attention using probe-dots that randomly appeared during MOT[12]. Crucially, electrophysiological enhancements were observed when the probe-dots were located exactly on the illusory line connecting two neighboring target items. This suggests that the tracked items are continuously maintained as a multi-item configuration in terms of a shortest-path polygon linking all items (Fig. 1b).

Here, we explicitly test the hypothesis that MOT with four target items is facilitated by an object-based mechanism, where a shortest-path polygon is maintained over time. To this end, we make use of the fact that, as the target items move, the shortest-path polygon has to undergo certain qualitative transitions, which we name here "topological" transitions (Fig. 1c): (i) The order in which the items are connected in the shortest-path polygon can suddenly switch, which we refer to as "flip". This reflects a discontinuous change of the polygon shape. Moreover, the polygon shape can switch either (ii) from convex to concave, or (iii) from concave to convex. We note that all three kinds of topological transitions correspond to the appearance and/or disappearance of curvature minima and maxima, which are known to be essential for shape processing[13–18]. If MOT relies on an object-based representation of an abstract multi-item configuration, we hypothesize that, at any given time during tracking, it will be possible to decode the likelihood of the occurrence of topological transitions from the ongoing electrophysiological signal of the subject performing the MOT task.

## Methods

### Subjects

Thirty-eight subjects (27 female, 11 male) performed in the current object-tracking task with a mean age of 26.921 (SD = 4.967) and received monetary compensation for their participation. Gender information was obtained by questionnaire. No information about socioeconomic status, or race/ethnicity was collected. Subjects provided written informed consent to participate in this study. All subjects were right-handed, had normal or corrected-to-normal vision, and did not report any psychological or neurological contraindications. The experiment was approved by the local ethics board of the Otto-von-Guericke University Magdeburg (no. 139/24). No preregistration was conducted for this study.

### Stimulus and procedure

The subjects had to perform in a modified multiple-object tracking task while their continuous EEG was concurrently recorded. All visual stimulation was presented on a 24" LED screen (Asus VG248QE) in white on a black background using PsychToolbox[19]. A small dot (0.11°) in the middle of the screen was present throughout the experiment, which the subjects had to fixate. At the start of each trial, eight visually indistinguishable hollow squares (0.45°) appeared on the screen, of which four started to blink twice by filling them white (200 ms) in order to designate them as targets for the current trial. Once all eight items were present as hollow squares again, they started moving around in a pseudorandom fashion in a designated field on the screen of 14.4° × 14.4°. Crucially, motion trajectories were calculated offline for each trial, with no trajectory occurring twice throughout the experiment, and item positions for each frame and trial being stored offline. The

## a) Behavioral probe response

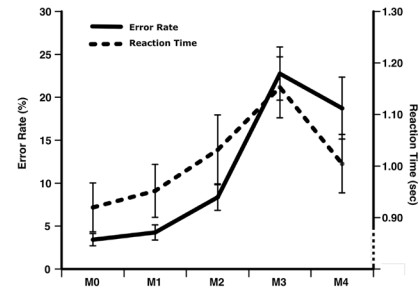

## b) Probe-related responses

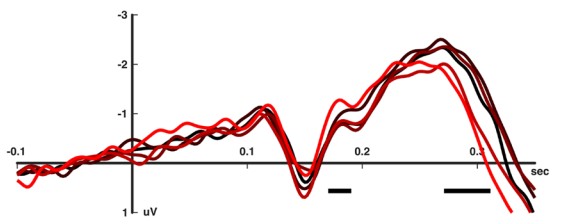

## c) Transition-related responses

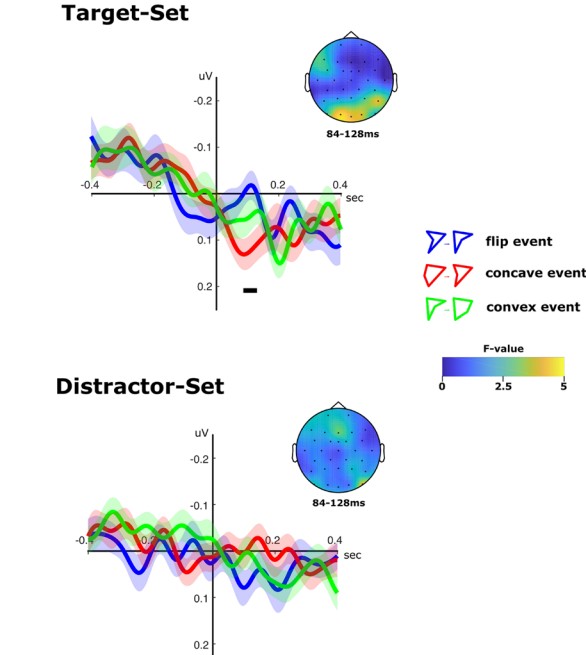

**Fig. 2 | Behavioral and event-related results. a** Increases in match between target items and probe items at the end of each trial were related to decreases in performance except for the full-match (M4) condition across all subjects (*N* = 38). **b** The N170-component towards the probe-display showed a distinct modulation for the full-match condition, while the N270-amplitude decreased with an increase in match between attended items and probed items. **c** The average event-related signal towards the three transition events during tracking showed a significant modulation around 84–128 ms after transition onset over occipital electrode sites. This modulation was only present for transition events of the illusory polygon within the target items, but not for the illusory polygon encompassing the distractor items.

algorithm for the calculation of the trajectories is described elsewhere in detail[9,20] but importantly, the inter-item distance between any two items never fall below (0.8°), or any item has moved outside the predefined field. Items moved with a constant speed of (4.1°/s) in a curved, continuous fashion, with directions changing randomly in non-predictable intervals for each item individually. Trajectories were calculated for individual items and later randomly combined to sets of eight trajectories to obtain the trials with the crucial condition for each combination in that no two items may approach below 0.8° in any frame. This strategy made interactions across items impossible while permitting random, non-predictable motion without occlusions. Motion lasted 4 s, after which four of the eight items again changed from an outlined square into a filled square. Those four filled squares constituted the probe and could correspond with the previously designated targets in five different ways. Either none (M0), one to three items (M1, M2, M3), or all of the probed items (M4) were identical with the target items. Subjects had to perform a two-alternative forced-choice task, indicating whether all four probe items matched the previously designated target items (M4) or not (M0, M1, M2, M3). Subjects used the index and middle fingers of the right hand for their response. After the response, the next trial started.

Subjects performed in 300 trials overall, including 60 trials per probe condition that were randomly shuffled. Participants were instructed to attend to the four target items throughout each trial without moving their gaze and respond as quickly and accurately as possible towards the visual probe at the end of each trial. All subjects were naïve as to the aim of the study. Crucially, no instructions or hints were given regarding specific tracking strategies. The illusory contour (discussed throughout the manuscript) was neither mentioned to the subjects nor was it ever physically present throughout the entire experiment.

### EEG-recordings

The electroencephalogram of each subject was recorded throughout the experiment using a 32-Channel actiCap system (Brain Products GmbH).

Impedances were kept below 10 k Ohm while the signal was recorded at 500 Hz and re-referenced offline to the mean of the left and right mastoid. All analyses were performed using FieldTrip[21] and custom Matlab scripts. The initial pre-processing was conducted as an artifact-identification and removal step and included a high-pass filter of 0.5 Hz to remove slow drifts. During this first step, the continuous signal was epoched ±1 s to the onset of each probe display. Those 300 epochs were then entered into an independent component analysis (ICA), at which point clear ocular and muscular components were identified in each subject and subsequently removed. The resulting dataset was used to calculate the event-related potentials time-locked to the visual probe for each of the five probe-match conditions (Fig. 2b).

A second artifact-corrected dataset was created for the initial continuous signal by back-projecting the unmixing matrix of the previous ICA, excluding the identified artifacts onto the individual channels of the initial continuous signal. For this continuous signal, additional temporal markers were calculated offline relating to three specific topological transition events repeatedly occurring throughout the motion phase.

Based on the calculated trajectories for each trial, abstract topological transition events were identified throughout the motion of the target items. First, at each time-point the four target locations were connected by the shortest-path polygon passing through all four items (Fig. 1b). Whenever the sequence of items in the shortest-path polygon changed, we define that the polygon performs a discontinuous 'flip' from one shape to another (Fig. 1c). The other two transitions are defined by the introduction or removal of a concavity from the polygon. To determine occurrences of these convex/concave transition events, we first calculated the convex hull of the set of target items (using the MATLAB functions *delaunayTriangulation* and *convexHull*). Mathematically, this convex hull always has between one and four corners. The cases of one or two corners are extreme exceptions (all four items coincide or lie on a line, respectively), which were ignored in the event detection. Thus, the convex hull always had either three corners,

implying the polygon connecting the four target items is concave, or four corners, implying the polygon is convex. Hence, if at two consecutive time frames, the number of corners increased from three to four, a "convex"-transition occurs (concave into convex shape), while a decrease from four to three defines a "concave"-transition (convex into concave shape). For the continuous artifact-corrected EEG signal, each of those three events was identified. Additionally, only transition events were included in the subsequent analyses that did not repeat within a 500 ms time window. Furthermore, any events occurring within 500 ms after motion onset or within 500 ms prior to the visual probe were excluded. Besides analyzing the electrophysiological signal related to the three transition events of the target set ("t_flip", "t_concave", "t_convex") throughout the motion phase, the very same events relating to the distractor set were analyzed as well ("d_flip", "d_concave", "d_convex"), serving as a cognitive baseline.

The signal was epoched time-locked to each of those six transition events over a time range of $-400$ to $+400$ ms and averaged.

### Correlation-analysis

The pre-processed EEG signal and the topological transition events in the illusory polygon were employed to determine the predictability of the individual events using event-related potentials. To this end, we started from the pre-processed EEG potentials $p_{sk}^n(t)$, where $t$ denotes time, $n = 1, \ldots, 32$ is the channel number $s = 1, \ldots, 38$ is the subject number, and $k = 1, \ldots, N_s$ is the number of the trial for that subject with $N_s$ being the total number of trials for subject $s$. Knowing the trajectories of all moving items for any trial $k$ of subject $s$, we computed the time point $t_{skei}$ of each topological transition event, where $e$ denotes the type of the topological transition (i.e. one of t_flip, t_convex, t_concave), and $i = 1, \ldots, N_{ske}$ is the event number, with $N_{ske}$ being the total number of topological events of type $e$ in trial $k$ of subject $s$. Based on this, we defined the time-locked event-related potential (ERP) $erp_{skei}^n(\tau)$ for channel $n$, subject $s$, trial $k$, event type $e$, and event number $i$, as

$$erp_{skei}^n(\tau) = p_{sk}^n(t_{skei} + \tau), \qquad (1)$$

for the relative time $\tau = \tau_{\min} \ldots \tau_{\max}$, where we set $\tau_{\min} = -400$ms and $\tau_{\max} = 400$ms unless stated otherwise. We first averaged the ERP over all subjects, trials, and events:

$$\overline{erp}_e^n(\tau) = \frac{1}{N_e} \sum_{s=1}^{38} \sum_{k=1}^{N_s} \sum_{i=1}^{N_{sk}} erp_{skei}^n(\tau), \qquad (2)$$

where $N_e = \sum_{s=1}^{38} \sum_{k=1}^{N_s} N_{ske}$ is the total number of events of type $e$ across all subjects and trials. Thus, $\overline{erp}_e^n(\tau)$, is the grand average signal for the three transition events. The average across $n = \{O1, Oz, O2\}$ is shown in Fig. 2c.

In order to reduce the dimensionality of the 32-channel signal, we performed a principal component analysis (PCA) of this grand average ERP, $\overline{erp}_e^n(\tau)$. To this end, we considered for each channel the time-dependent grand average ERP, $\overline{erp}_e^n(\tau)$, *for all transition event types $e$* (t_flip, t_convex, t_concave). The PCA resulted in a weight matrix $w_{cn}$, where $c = 1, \ldots, 32$ is the number of the principal component. For given principal component $c$, the weights $w_{cn}$ provide a topological scalp distribution describing how much the signal from any electrode $n$ contributes to the component $c$. Any grand average principle component is a linear combination of the grand average using the topological weight distribution $w_{cn}$:

$$\overline{erp}_e^c(\tau) = \sum_{n=1}^{32} w_{cn} \overline{erp}_e^n(\tau), \qquad (3)$$

Similarly, the individual ERPs and ongoing potentials corresponding to component $c$ are

$$erp_{skei}^c(\tau) = \sum_{n=1}^{32} w_{cn} erp_{skei}^n(\tau), \qquad (4)$$

and

$$p_{sk}^c(t) = \sum_{n=1}^{32} w_{cn} p_{sk}^n(t), \qquad (5)$$

respectively. The topological weight distributions across the scalp for the first four principal components are shown in Fig. 3a.

We next used the PCA weights to compute event signatures and an event-similarity measure (ESM). To this end, we followed a leave-one-out approach, for which we separated all 300 tracking trials into training and test trials. For any given subject $s$, we denote the set of all training trials $k$ by $K_{\mathrm{train}}^{sm}$ and the set of all test trials by $K_{\mathrm{test}}^{sm}$. Specifically, we define training and test trials depending on the matching condition, M0, …, M4, that has been probed at the end of the trial $k$: if the matching condition equals some previously fixed parameter $m = 0, \ldots, 4$, then the trial $k$ is a test trial, $K_{\mathrm{test}}^{sm}$. Conversely, if a matching condition different from $m$ was probed at the end of trial $k$, then it is a training trial, $K_{\mathrm{train}}^{sm}$. Note that since the match-condition was randomly determined at the *end* of each trial, it cannot affect any electrophysiological potential of that trial – it merely serves here as a random selector of approximately one-fifth of all trials. Thus, the parameter $m$ defined the way the trials are split into training and test runs.

For each subject $s$, each principal component $c$, and each split $m$, we next computed event-specific signatures by averaging the ERPs over all events $i$ of the training runs $k \in K_{\mathrm{train}}^{sm}$:

$$\overline{es}_{se}^{cm}(\tau) = \frac{1}{N_{\mathrm{train}}^{sem}} \sum_{k \in K_{\mathrm{train}}^{sm}} \sum_{i=1}^{N_{ske}} erp_{skei}^c(\tau), \qquad (6)$$

where $N_{\mathrm{train}}^{sem} = \sum_{k \in K_{\mathrm{train}}^{sm}} N_{ske}$ is the number of events of type $e$ in all training runs of subject $s$.

To obtain the ongoing ESM, we cross–correlate the ongoing potential $p_{sk}^c(t)$ of trial $k$ with the signature of the 'training' event $e$:

$$esm_{ske}^c(t) = \frac{\langle \Delta \overline{es}_{se}^{cm}(0, \tau) \Delta p_{sk}^c(t, \tau) \rangle_\tau}{\sqrt{\langle [\Delta \overline{es}_{se}^{cm}(0, \tau)]^2 \rangle_\tau \langle [\Delta p_{sk}^c(t, \tau)]^2 \rangle_\tau}}, \qquad (7)$$

where for some $\tau$-dependent quantity $x$, we introduced the average over the time interval from $\tau_{\min}$ to $\tau_{\max}$:

$$\langle x(\tau') \rangle_{\tau'} := \frac{1}{\tau_{\max} - \tau_{\min}} \int_{\tau_{\min}}^{\tau_{\max}} x(\tau') d\tau',$$

and we define the difference between quantity $x$ and its windowed average:

$$\Delta x(t, \tau) := x(t + \tau) - \langle x(t + \tau') \rangle_{\tau'}.$$

In other words, in our cross–correlation, we use the difference between the signal and its ongoing average over the time interval from $t + \tau_{\min}$ to $t + \tau_{\max}$. Moreover, we normalize the cross–correlation by the standard deviation of the event signature and the ongoing standard deviation of the potential, computed over the interval from $t + \tau_{\min}$ to $t + \tau_{\max}$.

To test whether and how the occurrence of some occurring 'test' events are reflected by the ESM, we next time-locked the ESM around all topological events of 'test' type $f$ over all test runs, $k \in K_{\mathrm{test}}^{sm}$, and then averaged (see Fig. 3b):

$$\overline{esm}_{sef}^{cm}(\tau) = \frac{1}{N_f^{=m}} \sum_{k \in K_{\mathrm{test}}^{sm}} \sum_{i=1}^{N_{skf}} esm_{ske}^{cm}(t_{skfi} + \tau), \qquad (8)$$

where $\tau = -\tau_w \ldots \tau_w$, with $\tau_w = 400$ms unless stated otherwise, and we introduced $N_{\mathrm{test}}^{sfm} = \sum_{k \in K_{\mathrm{test}}^{sm}} N_{skf}$, which corresponds to the number of occurrences of the test event type $f$ in all test runs for subject $s$.

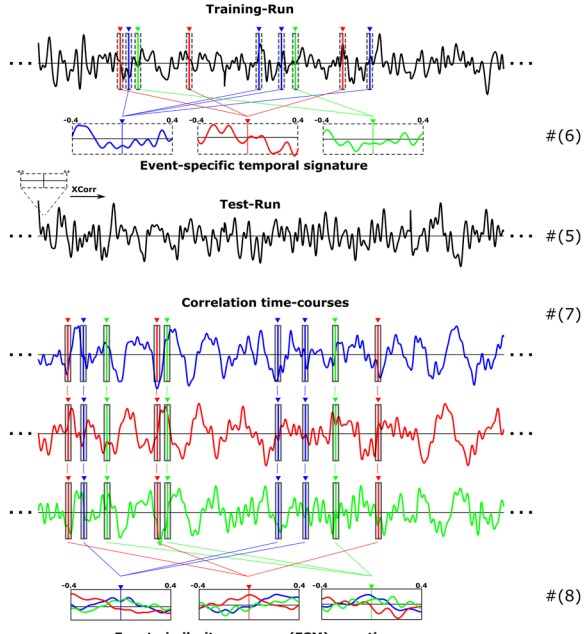

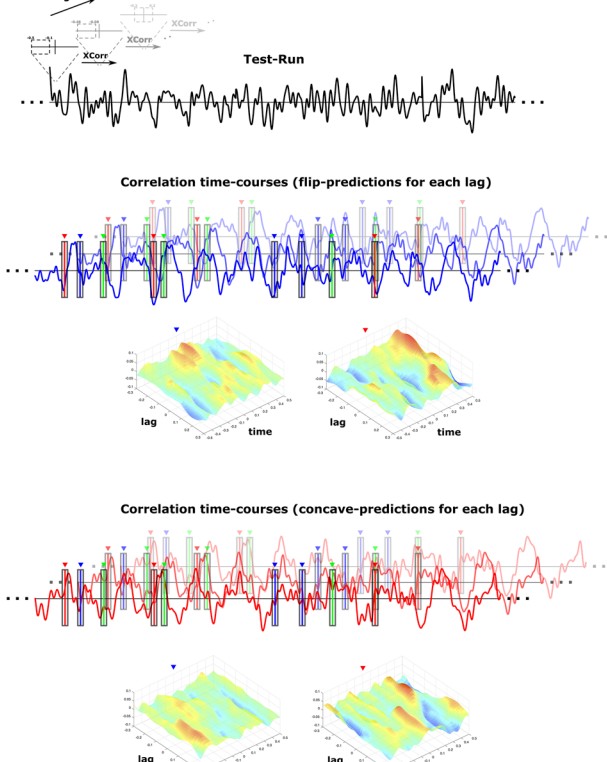

**Fig. 3 | Methods. a** A principle component analysis of the group-averages ($n = 38$) of the event-related signal for the three transition events was calculated to yield the first four factor load matrices of the average signal. **b** Each load matrix was used to calculate a projection of the original ongoing EEG-activity for each individual subject into one estimated component space. The individual continuous activity time courses for each of the first four principal components were analyzed using a correlation analysis. The event-specific electrophysiological signature around each transition point during tracking (±400 ms) was extracted and averaged for a set of four training runs and subsequently cross-correlated with the continuous signal of the fifth test run in a cross-validation approach. The resulting three correlation time-courses (one for each transition event) were averaged and yielded three event-similarity measures for each of the three transition events. **c** The same analysis was performed for cropped time-windows around the transition points (400 ms) with a systematic temporal shift relative to each transition onset from $-300$ to $+300$ ms in steps of 20 ms, resulting in two-dimensional event-similarity time-courses revealing not just when a transition is predicted to occur (time) but also based on which signal recorded relative to this event (lag).

We then performed a permutation correction. To this end, we first randomly shuffled the three event labels t_flip, t_convex, and t_concave among all training-run events, and we each time re-computed $\overline{esm}_{sef}^{cm}(\tau)$ following equations (4), (6-8), while keeping the initially computed PCA weight matrix. We denote the average of 50 so-obtained curves by $\overline{perm}_{sef}^{cm}(\tau)$, which we then subtract from the average time-locked ESM. Afterwards, we averaged over all five possible splits $m$ in training and test runs, and over all subjects:

$$\Delta \overline{esm}_{ef}^{c}(\tau) = \frac{1}{38} \sum_{s=1}^{38} \frac{1}{5} \sum_{m=0}^{4} \left( \overline{esm}_{sef}^{cm}(\tau) - \overline{perm}_{sef}^{cm}(\tau) \right), \quad (9)$$

For $c = 4$, the corrected average ESM, $\Delta \overline{esm}_{ef}^{c=4}(\tau)$, is plotted in Fig. 4a, where the individual plots and triangles correspond to the "test" event type $f$, and the color of each curve corresponds to the 'training' event type $e$. We also performed the same analysis Eqs. (1–9) using the distractor set events (d_flip, …) instead of the target set events (t_flip, …). However, we still used the same PCA weights $w_{cn}$ as before. The results for $c = 4$ are shown in Fig. 4b.

To probe which parts of the event-related potential contain the relevant information that allows to decode the event, we also varied the time interval, $[\tau_{min}, \tau_{max}]$ over which the event-related potential is defined, and repeated the computation of Eqs. (4), and (6–9) while keeping the initially computed PCA weight matrix. To vary the time interval, we fixed its length to $\tau_{max} - \tau_{min} = 400$ms, while we varied the center of the interval, $\bar{\tau} = \frac{\tau_{min} + \tau_{max}}{2}$, which we call time lag, between $\bar{\tau} = -300$ms … 300ms. In Eq. (8), we computed over a time interval with $\tau_w = 500$ms. For principal component $c = 4$, the resulting $\Delta \overline{esm}_{ef}^{c=4}(\tau, \bar{\tau})$ is plotted in Fig. 3c. It depends on $\tau$, which is time relative to the event, and the time lag $\bar{\tau}$, which is the center of the interval used to define the event signature, which is also used to compute the cross-correlation, Eq. (7).

**Statistical analysis**

Significant variations in reaction times and error rates were calculated using a one-factor five-level (five probe conditions) repeated measures analysis of variance (rANOVAs) design. The mean amplitudes for the event-related potentials elicited by the probe display were analyzed using one-factorial rANOVAs over the occipital electrodes (O9, Oz, O10) at two time-windows identified previously containing the N170 and N270 components[9,11].

The event-related potentials time-locked to the transition events of the target items were analyzed in an exploratory fashion and showed significant variations between the three events at occipital electrodes. The same one-factorial rANOVA was performed for the transition events of the distractor items.

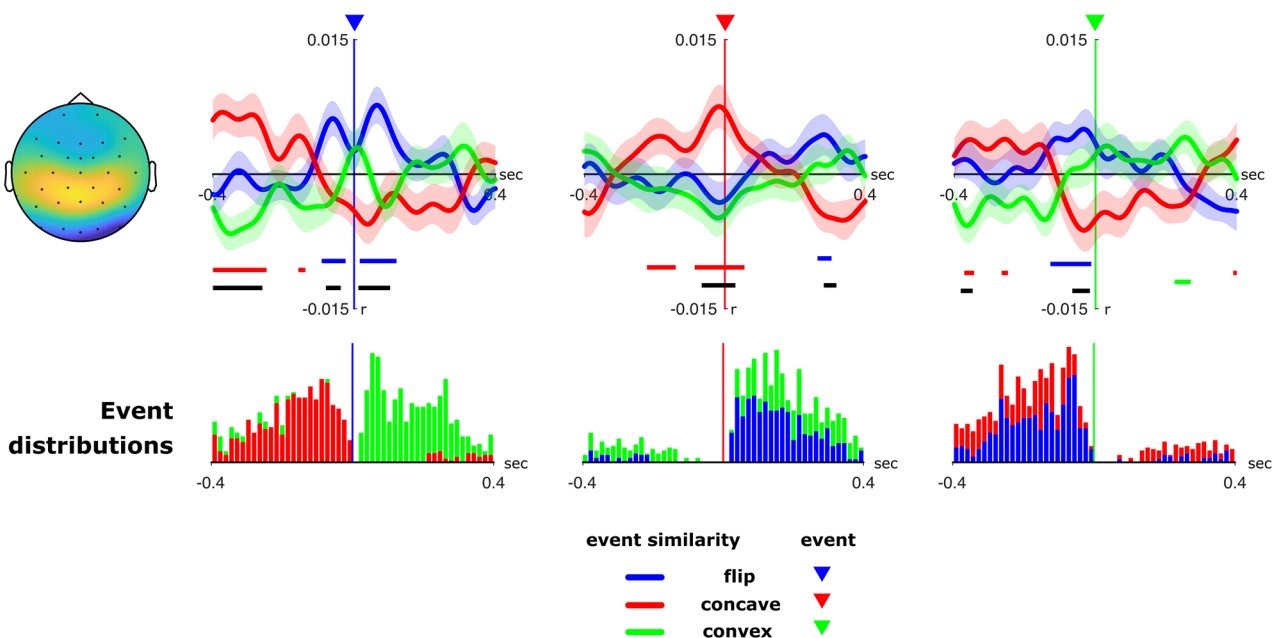

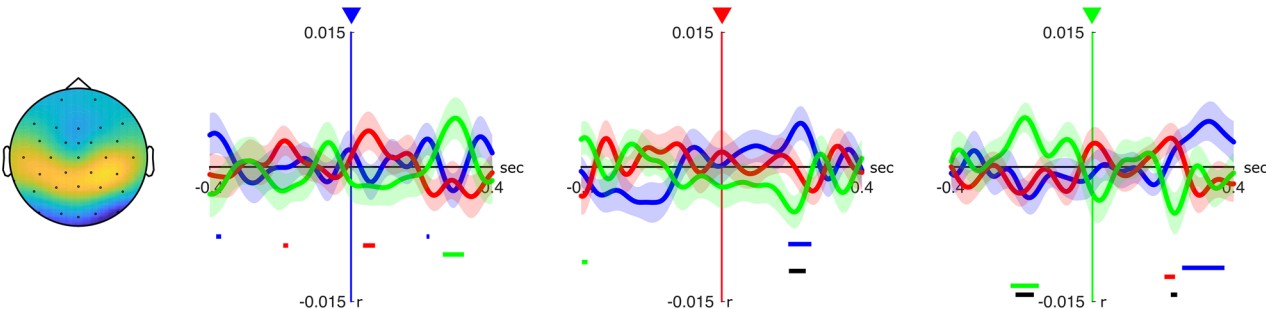

**Fig. 4 | Event similarity estimates.** Event similarity estimates for each transition event across all subjects ($n = 38$). **a** Time-courses of the likelihood of transition events in the target polygon occurring around each of the transitions based on the ongoing EEG-signal (component - C4). Triangles denote the time of each of the events occurring and horizontal colored lines indicate times at which event similarities are significantly larger than zero. Black horizontal lines indicate significant differences between all three event-similarity time-courses (one-factorial tANOVA with factors t_flip, t_concave and t_convex). The lower diagrams depict the stacked histogram of actual transition-events relative to each event-condition. **b** The same event-similarity measure related to the illusory polygon defined by the distractor set.

The correlation analyzes show that one particular continuous signal resulting from the back-projection of component 4 clearly carries information about target-event transitions. Subsequent analyzes focus on that component. Differences in the event-similarity measure from the correlation analysis between the three transition events for target items and distractor items were analyzed using one-factorial tANOVAs. Differences in the event-similarity measures between item-sets (targets/distractors) were investigated using two-factorial rANOVAs. The significance of the estimated likelihood of each event occurring at each time-point at an above-chance level was determined using one-sided t-tests. Further two-factorial rANOVAs were used to determine differences in event-similarity measures between two groups of high- and low-performing individuals, determined by the average error rate of the subjects. Multi-level rANOVAs were corrected if the assumption of sphericity was violated using the Greenhouse-Geisser epsilon. Corrected values are reported throughout the results section. All event-similarity measures at the three transition events (for the target set as well as the distractor set) were continuously scaled and exhibited normal distribution as determined by Shapiro–Wilk tests.

The parametric analysis results in 2-dimensional likelihood estimates across event-time and temporal lag for each event. Significant above-chance estimates were determined using one-sided t-tests corrected using cluster-based permutation.

### Reporting summary

Further information on research design is available in the Nature Portfolio Reporting Summary linked to this article.

## Results

### Event-related probe response

As in our previous MOT tasks, subjects were required to track four target items out of eight identical items during a movement phase. Afterwards, they were presented with four static probe items, of which either none (M0), some (M1-M3), or all four (M4) matched items in the target set. Subjects were then asked if the probe set fully matched the target set (M4). Consistent with our earlier findings[11], error rates in this binary decision and reaction times increased with the number of matching items (ER: $F(4,148) = 17.225$, $p < 0.001$, $\eta^2 = 0.318$, 90% CI [0.203, 0.393] RT: $F(4,148) = 16.998$, $p < 0.001$, $\eta^2 = 0.315$, 90% CI [0.201, 0.390]), except for the full-match condition (M4), for which subjects exhibited shorter reaction times ($t(37) = 4.751$, $p < 0.001$, $\eta^2 = 0.379$, 90% CI [0.172, 0.528]) and a

similar error rate to M3 ($t(37) = 0.775$, $p = 0.443$, $\eta^2 = 0.016$, 90% CI [0.000, 0.130]) (Fig. 2a).

The average electrophysiological response, time-locked to the appearance of the probe display, was also consistent with earlier findings. It contained the usual N170 and N270 components exhibiting previously observed amplitude modulations over occipital electrode sites (O1, Oz, O2) (Fig. 2b). The N170 mean amplitude differed among the match conditions (M0…M4) ($F(4,148) = 3.260$, $p = 0.020$, $\eta^2 = 0.081$, 90% CI [0.009, 0.137]), where the full-match condition showed a distinct negative enhancement compared to other partly matching conditions (M4 vs. M0: $t(37) = 3.087$, $p = 0.004$, $\eta^2 = 0.205$, 90% CI [0.0423, 0.373]; M4 vs. M3: $t(37) = 3.631$, $p < 0.001$, $\eta^2 = 0.263$, 90% CI [0.079, 0.427]), reflecting the visual discrimination of an object-based representation of the target set[9,11]. The N270 amplitude on the other hand increased with the mismatch between target and probe ($F(4,148) = 9.523$, $p < 0.001$, $\eta^2 = 0.205$, 90% CI [0.099, 0.279]; linear contrast: $F(1,37) = 15.005$, $p < 0.001$, $\eta^2 = 0.289$, 90% CI [0.097, 0.450]).

### Topological transitions elicit distinct electrophysiological responses

We wanted to test whether there are specific electrophysiological responses to topological transitions in the shortest-path polygon formed by the four target items. To this end, for each of the three kinds of topological transition (Fig. 1c) and for each electrode, we first calculated the mean responses, time-locked with a range of ±400 ms around all transition events across all trials and subjects. The occurrence of each transition event was thereby calculated from the known trajectories of the target item set (see Methods), where we denote the three kinds of topological transitions t_flip, t_concave, and t_convex (Fig. 1c). For these transitions, we obtained a total of 341, 376, and 370 events over all 300 trials, respectively. An exploratory analysis revealed that, strikingly, the event-related potential (ERP) at 84–128 ms relative to event onset across the occipital electrodes differed between the three transitions (O1, Oz, O2) ($F(2,74) = 4.482$, $p = 0.015$, $\eta^2 = 0.108$, 90% CI [0.013, 0.211]), where t_concave events elicited significantly larger positivities compared to t_flip events ($t(37) = 3.285$, $p = 0.002$, $\eta^2 = 0.226$, 90% CI [0.055, 0.393]) (Fig. 2c).

To provide a control, we note that the three topological transitions can also occur within the shortest-path polygon formed by the distractor items. We denote the corresponding transitions by d_flip, d_concave, and d_convex, respectively. If the observed differential response towards the topological transitions in the target set depends on an attentional mechanism, such a differential response should not appear with respect to the distractor set. Indeed, we find that ERPs evoked by the different topological events show a significant interaction effect with the item set (targets/distractors) ($F(2,74) = 5.057$, $p = 0.009$, $\eta^2 = 0.120$, 90% CI [0.019, 0.226]). The ERPs evoked by the different events do not show a significant variation within the distractor set ($F(2,74) = 1.356$, $p = 0.264$, $\eta^2 = 0.035$, 90% CI [0.000, 0.109]) (Fig. 2c).

### Topological events can be decoded from the ongoing electrophysiological signal

While we saw that the different topological transitions show distinct ERP responses, we were wondering whether, conversely, individual events could be decoded from the ongoing brain response. To this end, we used a correlation approach within each subject. First, to reduce the dimensionality of the data from the 32 channels, we performed a Principal Component Analysis (PCA) of the ERP signal, combined from all three transitions, each averaged over all target events and all subjects (see Methods). For the first four PCA components, this yielded the channel weight matrices (i.e., scalp distributions) shown in Fig. 3a. We then computed the projected EEG signal for each of these four components for each subject, and computed the mean ERP signal for the transitions in the target set. Each of these ERP signals serves as a distinct temporal signature for a given subject, given PCA component, and given topological transition (Fig. 3b). We used these signatures to decode the topological transition by cross–correlating the

**Table 1 | Event time course correlations**

| Relative Event | Prediction time course for event | Event distribution | Event distribution |
|---|---|---|---|
| | | correlation (r) | significance (p) |
| t_flip | t_concave | 0.389 | 0.005 |
| t_flip | t_convex | 0.352 | 0.012 |
| t_concave | t_flip | 0.471 | <0.001 |
| t_concave | t_convex | 0.358 | 0.011 |
| t_convex | t_flip | 0.351 | 0.013 |
| t_convex | t_concave | 0.502 | <0.001 |

Correlations between the distribution of actual transition events relative to each event and the event similarity time courses around each of those events.

signature with the entire ongoing projected EEG signal of that subject. The resulting correlation time-course represents the similarity of the signature with the current projected EEG signal, i.e., an event-similarity measure for the given topological transition. This cross–correlation was performed as a leave-one-out cross-validation; to obtain the temporal signatures, we used the average ERP signals from 4 out of 5 runs of the experiment, and we cross-correlated them with the ongoing projected EEG signal of the remaining run. Finally, to obtain a measure for the likelihood of the topological transitions, we averaged the continuous event-similarity measure (ESM) for each topological transition across the actual event occurrences (Fig. 3b).

Interestingly, we found a substantial modulation of the event-similarity measure around the topological transitions only for the C4 component of the PCA, while no information was found when using one of the first three components (C1-C3). The underlying scalp distribution (Fig. 3a) of C4 suggests bilateral ventral occipital sources.

To verify whether the topological transitions were correctly predicted, we first tested for ESM modulations in a small (±50 ms) interval around event occurrences (Fig. 4a). Indeed, the ESMs for the tree transitions differed significantly around t_flip events ($F(2,74) = 4.126$, $p = 0.020$, $\eta^2 = 0.100$, 90% CI [0.009, 0.202]) and t_concave-events ($F(2,74) = 5.613$, $p = 0.007$, $\eta^2 = 0.132$, 90% CI [0.025, 0.239]) with the ESM for t_flip being higher than chance (i.e. zero) at t_flip-events ($t(37) = 2.506$, $p = 0.009$, $\eta^2 = 0.145$, 90% CI [0.015, 0.312]) and the ESM for t_concave being higher than chance at t_concave events ($t(37) = 3.034$, $p = 0.002$, $\eta^2 = 0.199$, 90% CI [0.039, 0.367]). At t_convex events, the ESMs between all three transitions tended to differ ($F(2,74) = 3.006$, $p = 0.058$, $\eta^2 = 0.075$, 90% CI [0.000, 0.169]), however only the t_flip predictions showed a trend towards being higher than chance ($t(37) = 1.635$, $p = 0.055$, $\eta^2 = 0.067$, 90% CI [0.000, 0.2181]). Taken together, ESM time-courses were most distinct for t_flip and t_concave estimates.

Prediction functions relative to each event-condition did not just differ at time point 0 around each transition but showed distinct modulations over time (before and after transitions). For each transition, we computed the time intervals where each of the ESMs was significantly higher than chance (horizontal colored bars in Fig. 4a). For instance, we observed a higher-than-chance ESM of t_concave events 400–250 ms prior to t_flip events ($t(37) = 3.451$, $p < 0.001$, $\eta^2 = 0.243$, 90% CI [0.066, 0.409]). Likewise, t_flip ESM was significantly increased 250–300 ms after t_concave events ($t(37) = 1.873$, $p = 0.034$, $\eta^2 = 0.087$, 90% CI [0.000, 0.244]). Next, we compared the ESM time courses to the actual occurrences of transition events. Specifically, relative to each transition event, we calculated the distribution of any other event occurrences, yielding a temporal histogram (Fig. 4a—lower vertical bars). Indeed, we found that all average ESM time courses were highly correlated with the relative distribution of transition events (table 1).

As a control, we also tested for significant ESM modulations for the distractor set (Fig. 4b). To this end, we first computed the PCA decomposition based on the average ERP responses to the distractor events. The

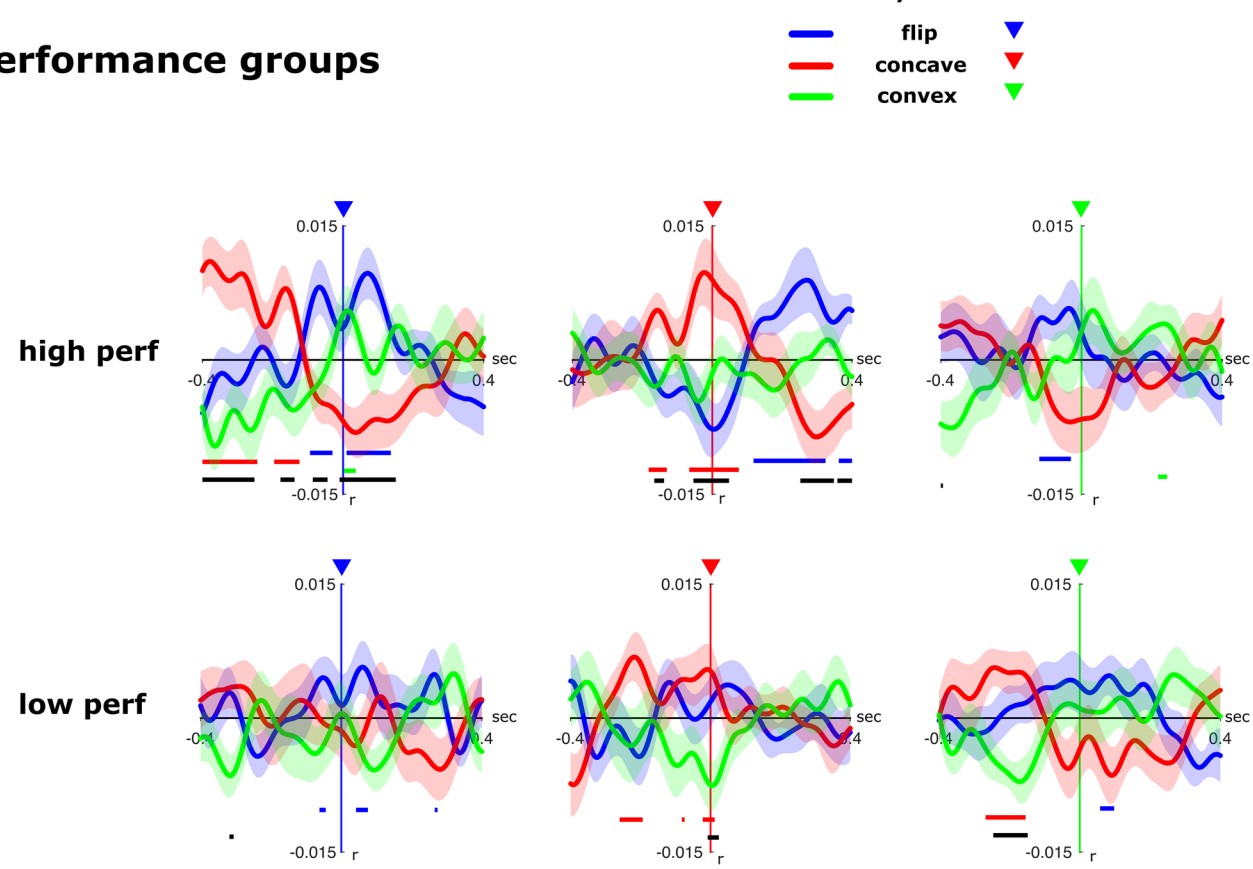

**Fig. 5 | Event similarity estimates for behavioral groups.** Two different groups of subjects, subdivided based on overall behavioral performance of the tracking task (high perf: $n = 19$; low perf: $n = 19$), exhibit different event-similarity time courses.

fourth component exhibited a very similar weight distribution as compared to the target-set PCA. However, the resulting ESM time courses did not show any systematic variation around the actual events (d_flip: $F(2,74) = 0.293$, $p = 0.747$, $\eta^2 = 0.008$, 90% CI [0.000, 0.0465]/d_concave: $F(2,74) = 0.388$, $p = 0.662$, $\eta^2 = 0.010$, 90% CI [0.000, 0.055]/d_convex: $F(2,74) = 0.438$, $p = 0.611$, $\eta^2 = 0.012$, 90% CI [0.000, 0.059]). This was in sharp contrast to the analysis performed on the target events. We directly compared the likelihoods for the three events at their time of occurrence (flip/concave/convex) across the item set (targets/distractors) using a two-factorial rANOVA which showed a significant main effect for the item set ($F(1,37) = 5.813$, $p = 0.021$, $\eta^2 = 0.136$, 90% CI [0.0114, 0.3015]), indicating that only the EEG response elicited by the target events reflect the topological shape transitions of illusory polygons connecting the attended items.

**Topological event predictability relates to subjects' tracking performance**

We also tested whether the maintenance of an illusory polygon throughout the motion phase correlates with the subjects' MOT performances. To this end, we divided subjects into two performance groups, based on overall accuracy in the matching task (high/low performance), and then compared the ESM-analysis-based prediction of topological transitions between these two groups. The ESM time courses averaged within each performance group are shown (Fig. 5). While the ESMs at ±50 ms around the individual target events failed to show significant interaction effects between transitions and performance groups (t_flip: $F(2,72) = 1.297$, $p = 0.280$, $\eta^2 = 0.034$, 90% CI [0.000, 0.107]/ t_concave: $F(2,72) = 2.488$, $p = 0.096$, $\eta^2 = 0.063$, 90% CI [0.000, 0.152]/ t_convex: $F(2,72) = 0.172$, $p = 0.833$, $\eta^2 = 0.005$, 90% CI [0.000, 0.032]), the ESM patterns, especially around t_concave events, hinted towards an enhanced event predictability for high-performance subjects. Indeed, the one-factorial rANOVAe for the t_flip and t_concave

events showed both a significantly higher ESM than chance and modulation between the ESMs of all transitions in the high-performance group (t_flip: $F(2,36) = 5.748$, $p = 0.007$, $\eta^2 = 0.242$, 90% CI [0.045, 0.393]/t_concave: $F(2,36) = 5.067$, $p = 0.017$, $\eta^2 = 0.220$, 90% CI [0.032, 0.371]), but not the low-performance group (t_flip: $F(2,36) = 0.646$, $p = 0.530$, $\eta^2 = 0.035$, 90% CI [0.000, 0.136]/t_concave: $F(2,36) = 2.973$, $p = 0.066$, $\eta^2 = 0.142$, 90% CI [0.000, 0.288]). These data at least suggest that a cognitive process maintaining an illusory configuration of target items enhances tracking performance.

**Temporally resolved target-event predictability**

The ESM analyzes so far showed that the EEG signal centered ±400 ms around t_flip and t_concave events allows to predict these events. Yet, we do not know which parts of the ±400 ms interval contain the information required for this prediction. To examine this, we cropped a 400 ms region from the temporal event-related signatures that we studied before, whose center position we varied relative to the event offset (time lag = −300 ms… +300 ms, in steps of 20 ms) (Fig. 3c). We then cross-correlated these cropped regions with the entire projected EEG signal of the C4 principal component and averaged around specific topological events to obtain ESM functions for each time lag (Fig. 3c).

Figure 6a shows such a 2-dimensional ESM function, illustrating the likelihood of t_flip events over a range of −500 ms to +500 ms relative to the t_flip-event, depending on the time lag. On visual inspection, t_flip events clearly are correctly predicted at their time of occurrence, even when using the smaller, cropped event-related signature. Importantly, this prediction is largest for time lags between −300 and −100 ms. Specifically, the ESM for the t_flip event at a time lag of −150 ms is significantly larger than chance at the time point of occurrence (±50 ms) ($t(37) = 2.192$, $p = 0.018$, $\eta^2 = 0.115$, 90% CI [0.005, 0.278]) (Fig. 6a). This points out that the

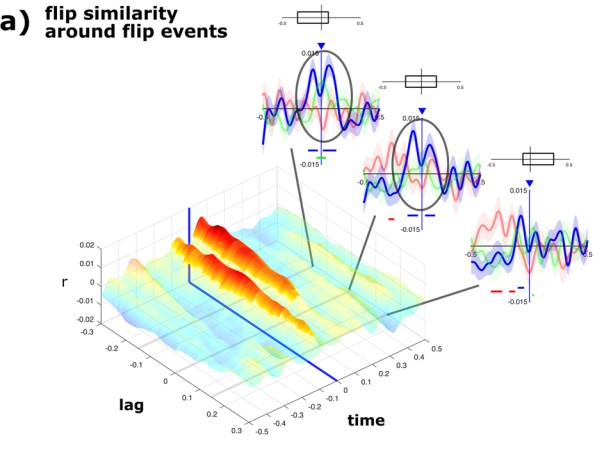

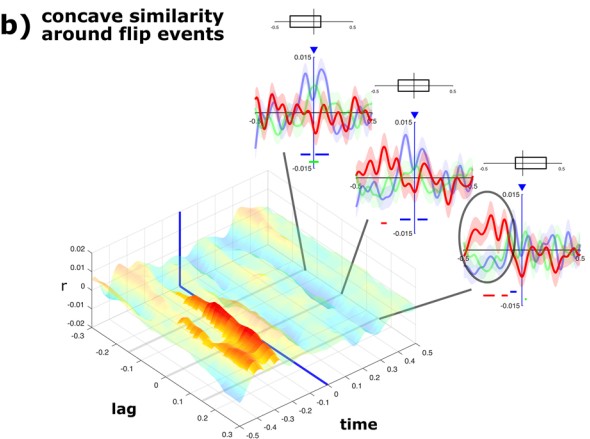

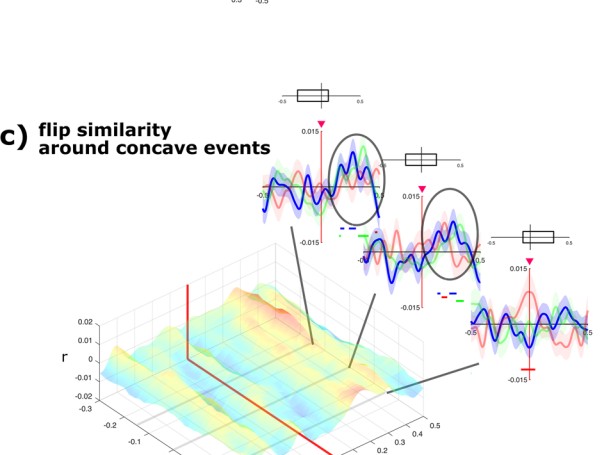

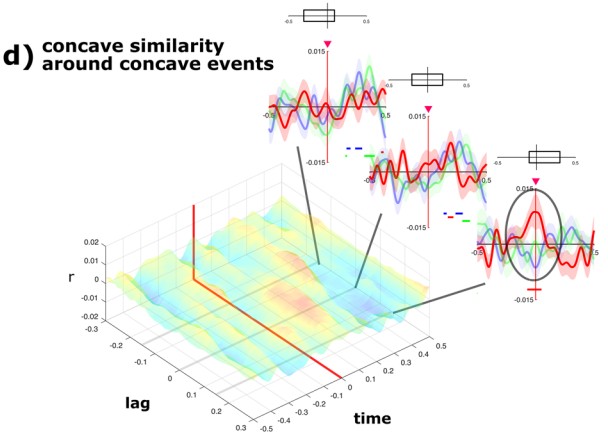

**Fig. 6 | Temporally resolved predictability functions.** The event-likelihoods are depicted for a cropped time-window of 400 ms systematically shifted from −300 ms to +300 ms (in steps of 20 ms) relative to each event, resulting in a two-dimensional similarity function showing when (time) an event is predicted to occur based on the temporally shifted (lag) EEG-signature relative to the event-onset. The shading of the function indicates statistical significance ($p < 0.05$) based on cluster-based permutation tests for all $r$-values (similarity) across time and lag parameters. One-dimensional similarity measures are presented for different lags (−150 ms, 0 ms, +150 ms). Above each of these one-dimensional similarity measures, the cropping information of the underlying EEG-signature relative to the event is indicated. Colored triangles and colored lines within the two-dimensional similarity function depict the time of occurrence of actual events ($n = 38$). **a** Estimated likelihood of t_flip-events occurring around actual t_flip-events are highest for signal occurring prior to the event. **b** Likelihood of t_concave-events is highest prior to t_flip-events. **c** T_flip-events are estimated to occur after actual t_concave-events. **d** Estimated likelihood of t_concave-events occurring around actual t_concave-events is highest for the signal occurring after the event.

information required to predict the t_flip events is embedded in the EEG signal *prior* to the event itself. Conversely, we find that the information required to predict t_concave events is encoded *after* the event itself (time lags +100 ms…+250 ms), as indicated by an increased ESM only for positive time lags (Fig. 6d). Specifically, the t_concave ESM at event occurrence (±50 ms) is significantly different from chance at a time lag of +150 ms ($t(37) = 2.215$, $p = 0.016$, $\eta^2 = 0.117$, 90% CI [0.005, 0.281]) (Fig. 6d—oval marker).

We also checked cross-predictions of topological events. For instance, in Fig. 6b, we show the cross–correlation of the cropped temporal event-related signatures of t_concave events with the EEG signal around t_flip events. Consistent with our earlier results, we find a higher ESM signature of t_concave occurrence 400 ms to 250 ms before t_flip events. However, this signature occurs only for positive time lag (+150 ms) (Fig. 6b – oval marker), confirming that information about t_concave events is mostly embedded after each t_concave event ($t(37) = 2.901$, $p = 0.003$, $\eta^2 = 0.185$, 90% CI [0.033, 0.354]). Similarly, Fig. 6c indicates a higher ESM signature of t_flip events after t_concave events (250–300 ms) only at a negative time lag of −150 ms ($t(37) = 2.407$, $p = 0.011$, $\eta^2 = 0.135$, 90% CI [0.011, 0.301]), confirming that information about t_flip events is mostly embedded before each t_flip event.

Taken together, we find that t_flip events are encoded in the EEG signal ~150 ms *prior* to the actual event (Fig. 6a–c), while t_concave events are encoded ~150 ms *after* the event (Fig. 6b–d). Note that performing the same statistical tests using the transition events of the distractor set does not reveal any systematic ESM patterns above chance.

## Discussion

Here, we show that the neural representation of items during Multiple-Object Tracking involves an illusory polygon, defined by the shortest path connecting all target items. Such a polygon morphs continuously throughout the tracking phase and thus exhibits different types of topological transitions. Here, we discussed three such transitions, of which two could be correctly decoded at their time of occurrence using the ongoing electrophysiological (EEG) signal. Specifically, we found that t_flip events, i.e., transitions in the order of the items within the shortest-path polygon, were encoded by an anticipatory EEG signal occurring *prior* to the transition. In contrast, the introduction of concavities into a convex shape induced EEG responses that *followed* the transition. Importantly, transition events could only be decoded if they occurred in the polygon formed by the target set, but not by the distractor set. Furthermore, the correct decoding of transition events was improved in subjects with a higher tracking task

performance. Taken together, our results confirm that the identified EEG signatures relate to an object-based attentional process.

The conscious attempt to track a number of moving items simultaneously as an abstract polygon has initially been shown to be behaviorally advantageous[8], which we later developed into an object-based tracking account[9]. Specifically, we have recently shown electrophysiological evidence for the maintenance of object-based representations during tracking[11,22]. The current data verify a critical prediction of this account, showing that topological transitions within these polygons can be decoded from the ongoing electrophysiological signal.

Our findings can also explain observations that have been taken to support location-based tracking accounts. For instance, tracking performance is known to be a function of the degree of spatial interference between the individual items[6,7,23], where interference was modulated by the variation of item number, item speed, or minimum item distances[2,5,7,24,25]. We note, however, that most of these spatial variations would also affect the frequency of transition events of an illusory polygon connecting the items, which will, in turn, increase the cognitive demand for maintaining this polygon. This suggests that, while location tracking is the main driver of object tracking, the correlation between tracking performance and spatial interference could —at least partially—be explained by an object-based account of tracking.

In the current data, we find a high degree of discriminability between topological transition events, in particular in the decoding of changes from convexity to concavity of an abstract shape. This implies that such types of transitions are associated with different underlying mechanisms and may exert different cognitive demands on the visual system.

We argue that transformations require a relatively little amount of resources when maintaining the convexity of the shape during morphing. However, this is not the case for the occurrence of concavities, which require mental transformations, whose processing time is known to increase with the degree of transformation[26,27]. Early accounts of object recognition highlight the importance of shape concavities in the segmentation and identification of objects. Hereby, shapes are parsed at contour locations at or close to global and local curvature minima of a closed contour (which include concavities)[14,18] and split into subunits. This notion has been supported by the argument that such curvature minima contain most shape information[13,16]. Concurrently, the visual system is more sensitive to local concavities compared to local convexities, as they have been shown to be perceived as more salient in change detection tasks[28,29] and visual search paradigms[30]. The enhanced decoding of the transition into concavities shown here is therefore well in line with the increased saliency shown in previous behavioral experiments.

Transition events may not just be discriminable by the visual system due to their inherent saliency, but also based on each transitions' impact on visual working memory load. This might provide further explanation for why we find distinct electrophysiological signatures specifically for t_flip and t_concave events. As mentioned before, transition events change the polygon's segmentation by introducing and removing local curvature minima and maxima. Such shape modulations would inadvertently affect the cognitive load on the visual system to continuously maintain representations of the shape. One way in which the ongoing cognitive load can be quantified in the visual domain is to record the contralateral delay activity (CDA)[31–34]. Hereby, the CDA amplitude does not necessarily index the number of singular features to be retained, but rather the number of conjunctions or objects maintained at any given moment[35,36]. Indeed, observations of the CDA show that changes in the segmentation of a real polygon can affect working memory load[37]. On the one hand, splitting a single non-morphing polygon into two polygons during a retention period is indicated by a roughly two-fold increase of the CDA amplitude[38]. This increase is, however, preceded by a sharp drop of the CDA, suggesting a so-called "reset"—a reestablishment of two entirely new representations in the visual system[39,40]. We propose that such a "reset" may also occur in morphing illusory polygons during our MOT experiments, when a previously convex polygon becomes concave. This would explain the high predictability of such t_concave events. On the other hand, splitting a non-morphing

polygon already separated by some other indicator merely 'updates' already established representations[37–40]. Similarly, in our morphing illusory polygons, the t_flip is characterized by local changes of already present concavities, suggesting a mere "update" of already present shape representations. Thus, t_flip and t_concave events likely trigger different working memory representations, i.e., they are functionally dissociated, which could explain the different electrophysiological responses to these two transition events. These ideas may also explain the high degree of confusion between t_flip and t_convex events. Both transitions do not increase the number of already present local concavities, and, therefore, do not require a "reset" of new working memory representations. Moreover, the reconsolidation of dropped representations ("reset") takes more time than an "update"[40]. This may help explain why in our MOT experiments, information about events in the electrophysiological signal appears later for t_concave events than for t_flip (Fig. 6). Taken together, the observed discriminability between the transition events is likely enhanced by differences in the respective induced changes in the working memory representations.

Our results suggest that transitions between concavity and convexity of a tracked shape imposes enhanced cognitive demands. We may ask what are the underlying neural mechanisms. It has been shown that abstract object representations in a tracking task are maintained within the ventral occipital cortex (V4/LO)[10,12]. In general, those areas seem to be sensitive to even subtle differences in global shape[41,42]. Additionally, electrophysiological results in primate V4 show that attending a morphing shape induces retinotopically localized gamma modulations[43]. Importantly, when task difficulty was modulated by increasing the speed of shape morphing, the amplitude of the gamma modulations increased as well, indicating an elevated neuronal engagement. Thus, the ventral occipital region is crucial during shape tracking, where the higher cognitive demands of the morph require additional resources. This evidence aligns with our results here, where the field distribution of the signal encoding the topological transitions (Fig. 4a) is consistent with neural sources in V4. Moreover, it aligns neatly with an enhanced decoding for cognitive demanding shape transitions of the polygon involving transitions in concavity and convexity.

It is likely that such transitions may involve modulations of activation patterns of specific neural subpopulations. Indeed, populations of neurons within ventral occipital areas show a high degree of contour selectivity, with a differential tuning towards the local curvature, i.e., whether it is locally concave or convex[44,45].

It is conceivable that any part of a morphing polygon might be maintained by the same population of broadly tuned units as long as this part keeps its local curvature (convex or concave). However, a qualitative curvature change would require a resource-demanding shift of the neuronal representation from a convexity-tuned population towards a concavity-tuned population and vice versa.

The present data provide evidence for the human visual system to support tracking of multiple locations by continuously maintaining an illusory polygon encompassing the target items. Critical transition events throughout the morph of that abstract polygon can be decoded using the ongoing electrophysiological response. Different electrophysiological signatures seem to reflect the involvement of different neural populations engaged in the encoding of local curvature. The data highlight the importance (i) of the continuous selection of visual shape information during tracking and (ii) of detecting concavities in visual shapes for accurate segmentation.

## Limitations

While tracking the illusory shape of a set of relevant items during motion proved beneficial in the current study, more research is needed to fully understand how such a process is applied in 3D real-world visual environments. Although 3D shapes are projection-invariant regarding their convexity, this does not need to be the case for concavities present in the contour of a three-dimensional morphing configuration. Further research needs to additionally address whether the employment of an illusory contour representation based on shortest paths generalizes for more than four

tracked. On the other end, illusory configurations with three items still carry relevant relational information about the tracked items, potentially improving tracking performance. However, shape transitions (i.e., concavities) as directly observed in the current study would not occur. Finally, it is worth emphasizing that while we focus on the formation and transformation of illusory shapes during multiple object tracking, other item-based mechanisms will significantly contribute to tracking performance. The present experiment does not allow us to assess how these mechanisms combine and interact during multiple object tracking. Future research is needed to address this question.

## Data availability

All data on which the results and conclusions are based on are available on the repository: https://osf.io/uga8d https://doi.org/10.17605/OSF.IO/UGA8D.

## Code availability

The custom scripts used for the analysis are available on the repository: https://osf.io/uga8d https://doi.org/10.17605/OSF.IO/UGA8D.

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

## Acknowledgements
This study was funded by the Deutsche Forschungsgemeinschaft Grant SFB-1436 B05. The funders had no role in study design, data collection and analysis, decision to publish or preparation of the manuscript.

## Author contributions
C.M.—conceptualization, data curation, formal analysis, investigation, methodology, writing—original draft. M.M.—formal analysis, methodology, validation, writing—review, and editing. J.M.H.—funding acquisition, resources, validation, writing—review, and editing. M.A.S.—funding acquisition, resources, validation, writing—review, and editing.

## Funding

## Competing interests
The authors declare no competing interests.
