## [Transparent Peer Review file · Communications Psychology]

Shape-transitions of a morphing illusory contour can be decoded during multiple-object tracking from the ongoing EEG

Corresponding Author: Dr Christian Merkel

Version 0:

Decision Letter:

Dear Dr Merkel,

Thank you for your patience during the peer-review process. Your manuscript titled "Decoding shape-transitions of a morphing illusory contour during multiple-object tracking from the ongoing EEG" has now been seen by 2 reviewers, and I include their comments at the end of this message. They find your work of interest but raised some important points. We are interested in the possibility of publishing your study in Communications Psychology, but would like to consider your responses to these concerns and assess a revised manuscript before we make a final decision on publication.

We therefore invite you to revise and resubmit your manuscript, along with a point-by-point response to the reviewers. Please highlight all changes in the manuscript text file.

Editorially, we consider it crucial that you situate the current study in the literature by clarifying how the work confirms, extends, or contradicts previous work. Please also make sure that the revised manuscript demonstrates that the experimental design is appropriate for the major conclusions of the study, and that alternative explanations are accounted for.

As you revise the manuscript in response to these issues, please also implement all requests in the attached Mandatory Revision Requests document. All requirements listed in this document need to be fully met, or the work will be returned to you for further revisions without peer review. This workflow is in place to increase the likelihood that the paper will be accepted for publication. It reduces the number of rounds of revision (and review) and ensures that the reviewers vet a version of the article that is compliant with journal policies. If you have any questions regarding the required revisions, please contact the journal prior to resubmission to avoid a negative outcome.

Please submit the following items:

- Revised manuscript
- Point-by-point response to the referees' comments
- Mandatory Revision Requests Table (attached).
- Cover letter (as a separate document)

via this link: Link Redacted .

** This url links to your confidential home page and associated information about manuscripts you may have submitted or are reviewing for us. If you wish to forward this email to co-authors, please delete the link to your homepage first **

Best regards,

Troy Lui, on behalf of

Shao-Min (Sean) Hung

Troy Lui, PhD
Associate Editor
Communications Psychology

Shao-Min (Sean) Hung, PhD
Editorial Board Member
Communications Psychology
orcid.org/0000-0002-8908-1497

REVIEWER EXPERTISE:

Reviewer #1: attention, visual perception

Reviewer #2: EEG, attention, visual perception

REVIEWER REPORTS:

Reviewer #1 (Remarks to the Author):

1. The results are generally straightforward, and I appreciate the analyses that went into being able to extract changes in the signal *during* MOT itself, and across the transitions over time. The coolest part of this paper is the demonstration that invisible topological structures are represented and that transitions from one topology to another can be decoded from EEG signals. But I'm not sure whether the resulting claim from these results (i.e., that MOT relies on a configuration mechanism) is warranted:

2. First, in terms of what we know about MOT. The current paper explores an interesting hypothesis that instead of attention being allocated to *individual* objects during tracking, it may be allocated to the *group* of objects — where the system tracks the abstract shape enclosing the objects. I appreciate the motivation of finite cognitive resources, but I was left confused about the need for this additional mechanism to the theory. We already know from many multiple-object tracking studies that the system is limited, such that we can only track 4-6 objects, which aligns with other ways we understand attention and working memory (i.e., the magic number 5 +/- 2). We also know that this process adaptively reallocates attention depending on the relationship between the targets and the distractors (see especially Belledonne et al., 2025, Psychological Review). If the main question is about topological structure being a critical way the system gets around limited resources, the current study has to show that adding a topological structure in fact helps with this resource allocation.

3a. Moreover, the method may also not be directly testing what the authors actually want to test. It's unclear whether these abstract shape configurations arise spontaneously (i.e., the system always will try to find the closest path connecting objects), versus the particular set-up of the experiment facilitating this. I worry that the way the experiment is set-up (see especially Fig. 1b) is such that there are probably other cues such as common motion that is giving rise to this abstract shape configuration. And we know that when object motions are constrained by their 'connectedness', even if this is invisible, multiple-object tracking is modulated (see Franconeri/Alvarez, Scholl).

3b. And even if this abstract shape configuration is represented spontaneously, objects in the real-world don't often follow a closed abstract shape configuration. So how would tracking play out then, when objects do not form a closed shape? For instance, objects that are laid out in a line, or in a 3D world, will not necessarily be easily configured into the nice concave/convex shapes that they have. Thus, I think the results may be a product of the particular design, but it's unclear whether this would really be a *general* feature of MOT.

4. The idea (that topological relations between elements are extracted) is not necessarily new (see Yousif & Brannon, 2025), but it is interesting that transitions from one topology to another might also lead to differences in the EEG signal. I think this is

the main contribution of the paper. Thus, the paper will benefit from refocusing the claims to be about the impact of transitions — especially the resetting/updating mechanism that they describe in the general discussion. The existing paradigm seems less about what MOT relies on, and more what happens when grouped objects transition from one configuration to another.

Reviewer #2 (Remarks to the Author):

Review of manuscript

Merkel, C., Merkel, M., Hopf, J-M., Schoenfeld, M.A. Decoding shape-transitions of a morphing illusory contour during multiple-object tracking from the ongoing EEG

Summary

The current ms reports a study that used EEG and a novel decoding approach to assess the proposal that performance in multi-object tracking (MOT) tasks depends, at least in part, on an object-based mechanism. Participants completed an MOT task that included three types of transformations in the shape of the illusory polygon formed by the imaginary line connecting the four target items: 'flips,' in which target motion causes a vertex to switch from one side of a polygon to the other; a switch from a convex to a concave shape, and; a switch from a concave to a convex shape. These topological transition events elicited unique ERP signatures, in keeping with previous findings, and both flip events and transitions from convex to concave shapes could be decoded from the ongoing EEG signal. The authors discuss potential neural correlates of these effects and argue that the results support an object-based process in MOT.

General Assessment

The paper is well written for the most part and includes a useful introduction to the MOT paradigm and studies of relevance to location- versus object-based processes. Methods and Results are likewise clear and appropriate to the stated aims of the study. I appreciated the inclusion of control analyses looking at ERPs and decoding performance focused on topological transition events for distractors items, which increases confidence in the reported results. I have not serious issues with the article in its present form and I think it makes a nice addition to the author's previous work in this area and will make a valuable contribution to the MOT literature. I do have a few minor suggestions, which I include below.

Minor Comments (in order of appearance in ms)

1. Sentence beginning at end of Line 81. This sentence is kind of confusing, to me. Consider revising for clarity.
2. Sentence beginning at end of Line 84. Consider rewording to, something like, "Strikingly, when all possible target items were included in the probe set, a distinct..., compared to conditions where only a subset of targets were included." You might also consider revising your earlier description of the standard MOT task to highlight that, in some cases, only a subset of target items are presented as probes at test.
3. Sentence on line 103 that includes "we term this event as 'flip'." Consider rephrasing as "which we refer to as a 'flip'."
4. First sentence on Line 121 beginning with "As for our previous MOT tasks". Consider rephrasing as "As in our previous MOT tasks, .."

End

Version 1:

Decision Letter:

Dear Dr Merkel,

Your manuscript titled "Shape-transitions of a morphing illusory contour can be decoded during multiple-object tracking from the ongoing EEG" has now been seen by our reviewers, whose comments appear below. In light of their advice I am delighted to say that we are happy, in principle, to publish a suitably revised version in Communications Psychology.

We therefore invite you to revise your paper one last time to address the remaining concerns of our reviewers and a list of editorial requests. At the same time we ask that you edit your manuscript to comply with our format requirements and to maximise the accessibility and therefore the impact of your work.

EDITORIAL REQUESTS:

SUBMISSION INFORMATION:

OPEN ACCESS:

*** TRANSPARENT PEER REVIEW:** Communications Psychology uses a transparent peer review system. On author request, confidential information and data can be removed from the published reviewer reports and rebuttal letters prior to publication. If you are concerned about the release of confidential data, please let us know specifically what information you would like to have removed. Please note that we cannot incorporate redactions for any other reasons.

*** CODE AVAILABILITY:** All Communications Psychology manuscripts must include a section titled "Code Availability" at the end of the methods section. We require that the custom analysis code supporting your conclusions is made available in a publicly accessible repository at this stage; please choose a repository that generates a digital object identifier (DOI) for the code; the link to the repository and the DOI must be included in the Code Availability statement. Publication as Supplementary Information will not suffice.

*** DATA AVAILABILITY:**

Link Redacted

Best regards,

Troby Lui

Troby Lui, PhD
Associate Editor
Communications Psychology

Shao-Min (Sean) Hung, PhD
Editorial Board Member
Communications Psychology
orcid.org/0000-0002-8908-1497

REVIEWERS' COMMENTS:

Reviewer #1 (Remarks to the Author):

I thought drawing the emphasis to 'decoding' topologies helped clarify the paper's contribution, and the new limitations section addressed my concern about how this would actually play out in reality (which posing a new interesting question about the constraints of vision given a 3D world!). I thank the authors for addressing my different concerns.

Reviewer #2 (Remarks to the Author):

The authors have adequately addressed each of my concerns.

Reviewer #1 (Remarks to the Author):

*1. The results are generally straightforward, and I appreciate the analyses that went into being able to extract changes in the signal *during* MOT itself, and across the transitions over time. The coolest part of this paper is the demonstration that invisible topological structures are represented and that transitions from one topology to another can be decoded from EEG signals. But I'm not sure whether the resulting claim from these results (i.e., that MOT relies on a configuration mechanism) is warranted:*

We thank the reviewer for acknowledging the novelty of the present data and appreciate the opportunity to elaborate on the conclusions regarding object tracking and visual grouping.

*2. First, in terms of what we know about MOT. The current paper explores an interesting hypothesis that instead of attention being allocated to *individual* objects during tracking, it may be allocated to the *group* of objects — where the system tracks the abstract shape enclosing the objects. I appreciate the motivation of finite cognitive resources, but I was left confused about the need for this additional mechanism to the theory. We already know from many multiple-object tracking studies that the system is limited, such that we can only track 4-6 objects, which aligns with other ways we understand attention and working memory (i.e., the magic number 5 +/- 2). We also know that this process adaptively reallocates attention depending on the relationship between the targets and the distractors (see especially Belledonne et al., 2025, Psychological Review). If the main question is about topological structure being a critical way the system gets around limited resources, the current study has to show that adding a topological structure in fact helps with this resource allocation.*

The reviewer brings up an important point about the fundamental idea on configuration tracking. In the last 20 years, several studies have provided evidence for individual location tracking explaining quite well the inter- and intrasubject variability of multiple-object tracking behavior in humans. Most importantly, the concept of configuration tracking does not dispute the findings of that research. The idea of a mechanism operating on an abstract configuration during tracking is rather making an additional contribution to standing theories of tracking (FINSTs, FLEXs etc.) than replacing them. In fact, the behavioral data from our own tracking-papers (2015 to 2025) consistently shows a parametric relationship between the number of probes matching with the number of relevant targets and subjects' reaction times and error rates, clearly suggesting an underlying cognitive resource shared amongst individual tracked locations in line with Pylyshyn, Cavanagh, Alvarez, Horowitz etc.

What our research attempts to contribute is a mechanism explaining 'gaps' in the location tracking accounts. The classic study from Yantis (1992) shows that direct cueing of configuration information can be used by subject to improve tracking. This information must be represented in some way. Thus, we do not ask, *whether* configurational information is maintained but rather *how* it is maintained by the system. Our usual probe-all tracking-task attempts to cue this configuration information indirectly right after motion ceases and shows that probing all target items simultaneously provides considerable behavioral advantage over probing any other combination of target and distractor items. Pure location-tracking cannot account for these results. In previous work (Merkel et al. 2014, 2015, 2017, 2022) we have provided convincing evidence for the existence of both location and configuration tracking.

We now expand on the motivation for the study in the introduction of the revised manuscript to make this clear.

3a. Moreover, the method may also not be directly testing what the authors actually want to test. It's unclear whether these abstract shape configurations arise spontaneously (i.e., the

system always will try to find the closest path connecting objects), versus the particular set-up of the experiment facilitating this. I worry that the way the experiment is set-up (see especially Fig. 1b) is such that there are probably other cues such as common motion that is giving rise to this abstract shape configuration. And we know that when object motions are constrained by their 'connectedness', even if this is invisible, multiple-object tracking is modulated (see Franconeri/Alvarez, Scholl).

The reviewer is absolutely right that grouping can facilitate behavior during object tracking, including common motion (Howe, 2010). Furthermore, object extrapolation has been observed in moving objects for displays up to two relevant objects (Frensch, 2007; Horowitz, 2010). However, for more than two simultaneously tracked objects, behavioral advantages of extrapolation strategies diminish considerably (Zhong, 2014; Howe, 2012).

It is important to note that, in the current study, none of the above-mentioned properties apply to our stimulus design:

First, trajectories were calculated for individual items only (irrespective of any other items' trajectories in a prospective trial). Consequentially, items within a trial do not 'interact'. There is no deterministic law prohibiting occlusions within a trial like 'bouncing off' or other 'force fields' like in other studies. Items were instead combined within a trial by random selection with the constraint, that at no point during motion items would be allowed to approach below 0.8° . Thus, although occlusion is impossible, there is no way to predict, how two approaching items would avoid such an event.

In addition, individual motion trajectories are strictly non-linear but trace in a curved manner changing in magnitude and direction at random intervals (individually for each item).

These design elements ensure a non-predictable behavior of the motion history within the items and between items.

We included these additional design information in the methods section of the revised manuscript.

*3b. And even if this abstract shape configuration is represented spontaneously, objects in the real-world don't often follow a closed abstract shape configuration. So how would tracking play out then, when objects do not form a closed shape? For instance, objects that are laid out in a line, or in a 3D world, will not necessarily be easily configured into the nice concave/convex shapes that they have. Thus, I think the results may be a product of the particular design, but it's unclear whether this would really be a *general* feature of MOT.*

We thank the reviewer for bringing up this interesting question: How does tracking within an artificial environment compare to tracking within a real-world 3D environment? Liu (2005) suggests that in real-world applications, tracking is performed within an allocentric reference frame with rotations of that frame having no impact of relative motion within the tracking space. The question about abstract shapes being used to facilitate tracking in 3D-environments goes beyond the motivation of the present study. Nevertheless, following Liu's idea of tracking being performed in an 3D allocentric space, illusory contours could connect target-locations within the real world as well. The special case mentioned by the reviewer (more than 2 points on a straight line) is hereby less likely to occur than within the 2D retinal space. Interestingly, convexity as a property of the illusory 3D-shape is projection-invariant across transformations, thus its 2D projection would retain a convex curvature at each location. In that case curvature might be especially beneficial in maintaining location information within 3D allocentric space. We now present a short introduction of this idea in the limitations of the study.

4. The idea (that topological relations between elements are extracted) is not necessarily new (see Yousif & Brannon, 2025), but it is interesting that transitions from one topology to another

might also lead to differences in the EEG signal. I think this is the main contribution of the paper. Thus, the paper will benefit from refocusing the claims to be about the impact of transitions — especially the resetting/updating mechanism that they describe in the general discussion. The existing paradigm seems less about what MOT relies on, and more what happens when grouped objects transition from one configuration to another.

We thank the reviewer for this comment and the suggestion for a shift in the general discussion. Please note, that we do provide evidence for a direct impact of the transition events on multiple object tracking performance. As response also to point 2 of the reviewer, we will stress the point, that the process of maintaining illusory shape information in the current paper is connected to its behavioral relevance:

- (1) Subjects in which contour transformations can be decoded more accurately do show a better tracking performance.
- (2) Contour transformations can only be decoded within the target set and not the distractor set.

Thus, shape information is not just passively ‘dragged along’ but is specific for the target items and modulates the behavioral responses at the end of the tracking phase.

Reviewer #2 (Remarks to the Author):

The current ms reports a study that used EEG and a novel decoding approach to assess the proposal that performance in multi-object tracking (MOT) tasks depends, at least in part, on an object-based mechanism. Participants completed an MOT task that included three types of transformations in the shape of the illusory polygon formed by the imaginary line connecting the four target items: ‘flips,’ in which target motion causes a vertex to switch from one side of a polygon to the other; a switch from a convex to a concave shape, and; a switch from a concave to a convex shape. These topological transition events elicited unique ERP signatures, in keeping with previous findings, and both flip events and transitions from convex to concave shapes could be decoded from the ongoing EEG signal. The authors discuss potential neural correlates of these effects and argue that the results support an object-based process in MOT.

General Assessment

The paper is well written for the most part and includes a useful introduction to the MOT paradigm and studies of relevance to location- versus object-based processes. Methods and Results are likewise clear and appropriate to the stated aims of the study. I appreciated the inclusion of control analyses looking at ERPs and decoding performance focused on topological transition events for distractor items, which increases confidence in the reported results. I have no serious issues with the article in its present form and I think it makes a nice addition to the author’s previous work in this area and will make a valuable contribution to the MOT literature. I do have a few minor suggestions, which I include below.

We thank the reviewer for this kind assessment of our work and adopted the following suggestions in the manuscript. These changes do improve the legibility of these passages considerably.

Minor Comments (in order of appearance in ms)

1. Sentence beginning at end of Line 81. This sentence is kind of confusing, to me. Consider revising for clarity.

2. Sentence beginning at end of Line 84. Consider rewording to, something like, “Strikingly, when all possible target items were included in the probe set, a distinct..., compared to conditions where only a subset of targets were included.” You might also consider revising your earlier description of the standard MOT task to highlight that, in some cases, only a subset of target items are presented as probes at test.
3. Sentence on line 103 that includes “we term this event as ‘flip’.” Consider rephrasing as “which we refer to as a ‘flip’.”
4. First sentence on Line 121 beginning with “As for our previous MOT tasks”. Consider rephrasing as “As in our previous MOT tasks, ..”